# Pseudo-Non-Linear Data Augmentation via Energy Minimization

## Abstract

We propose a novel and interpretable *data augmentation* method based on *energy-based modeling* and principles from *information geometry*. Unlike black-box generative models, which rely on deep neural networks, our approach replaces these non-interpretable transformations with explicit, theoretically grounded ones, ensuring interpretability and strong guarantees such as energy minimization. Central to our method is the introduction of the *backward projection* algorithm, which reverses dimension reduction to generate new data. Empirical results demonstrate that our method achieves competitive performance with black-box generative models while offering greater transparency and interpretability.

## 1 Introduction

*Data augmentation* has advanced significantly in recent years, primarily due to the increasing use of generative models to meet the growing demand for large datasets (Feng et al., 2021; Wong et al., 2016). Despite their success, these generative models often rely on modern deep neural networks, which are typically treated as black boxes, raising concerns about their interpretability (Guidotti et al., 2018). For instance, the popular *autoencoder* model encodes original data into a compact latent representation and then decodes it back, with both processes usually handled by black-box neural networks (Kingma & Welling, 2022). Consequently, even when these models perform well, the lack of understanding of the underlying transformations makes it difficult to control the generated outputs, forcing researchers to depend heavily on empirical heuristics.

A natural approach to developing a more interpretable data augmentation method is to replace black-box transformations with more explicit ones (Rudin, 2019). In this work, we take inspiration from the autoencoder model, which consists of *encoder* and *decoder*. Encoder, when viewed as a form of *dimension reduction* (Wang et al., 2016), contributes to the model's success by acting as a form of regularization and potentially avoiding sparsity through encoding data into a low-dimensional latent representation space. Indeed, various data augmentation methods adopt this philosophy, where compact representations are first learned via neural networks, incorporating dimension reduction as a key component of the pipeline (Maharana et al., 2022). However, while dimension reduction is a well-established field in data science, two main obstacles prevent its direct application to data augmentation. First, classical methods like Principal Component Analysis (Wold et al., 1987) and Singular Value Decomposition (Stewart, 1993) inherently rely on *linear* projection in the ambient space (e.g., Euclidean), making the straightforward application of these methods unsuitable for certain modality such as images. Second, the decoder—which aims to reverse the dimension reduction to generate new data—is highly non-trivial to design even for these classical linear methods. This is one of the main reasons why modern generative models rely on black-box transformations.

We address both issues by proposing a new framework and a data-centric algorithm. The framework introduces non-linearity through the well-known *energy-based model* (Xie et al., 2016), and is built upon recent developments in the *log-linear model on partially ordered sets (posets)* (Sugiyama et al., 2016; 2017) and *information geometry* (Amari, 2016; Amari & Nagaoka, 2000; Ay et al., 2017): the log-linear model on posets embeds structured data (e.g., tensors) as discrete probability distributions via an explicit mapping $\varphi$ into a statistical manifold $\mathcal{S}$, and subsequently, provides intricate geometric structure of the data that enables efficient dimension reduction method via projection in $\mathcal{S}$. Building on this, the proposed algorithm, termed *backward projection*, aims to reverse this *forward* projection process to generate new data via projection again. The core idea of backward projection is simple and general: given a new point in the low-dimensional latent representation

space, we identify its $k$-nearest latent representations of the original data (obtained via forward projection) and use them to create a target subspace to project *backward* onto. A key insight of the proposal is its ability to exploit the interplay between linearity and non-linearity of projection: the linearity arises from the *divergence minimizing* property when projecting onto *flat* low-dimensional sub-manifolds defined by *linear* constraints on the coordinate systems provided by the log-linear model on posets; however, these projections are inherently non-linear as the space $\mathcal{S}$ is curved. This interplay leads to what we refer to as *pseudo-non-linear* data augmentation.

By combining backward projection with the log-linear model on posets, our approach benefits from explicit, energy-based transformations: these non-linear projections are interpretable, fully white-box, and energy-minimizing, while the framework offers the potential to capture intricate information beyond the ambient space structure. Our contributions are summarized as follows:

- We introduce a novel framework for modeling structured data (e.g., tensors) within a statistical manifold via energy-based modeling. Unlike previous works on information geometry, which focused on a single probability distribution, we consider multiple distributions simultaneously, offering a "meta" learning perspective that may be of independent interest.

- We propose the *backward projection* algorithm, a data-centric method that reverses dimension reduction, which we then utilize to develop a novel data augmentation method within our framework.

- We demonstrate the effectiveness of the proposed data augmentation method. Results show that our approach achieves competitive performance compared to black-box generative models such as autoencoder through simple, transparent, and interpretable algorithms, underscoring its interpretability.

## 2 RELATED WORK

### 2.1 DATA AUGMENTATION

**In the Era of Deep Generative Models.** Data augmentation has proven to be highly effective in enhancing deep learning training by increasing dataset size, improving model robustness (Rebuffi et al., 2021), and introducing implicit regularization (Hernández-García & König, 2018). These techniques have been applied across various modalities, including text (Shorten et al., 2021; Feng et al., 2021; Li et al., 2022a) and images (Shorten & Khoshgoftaar, 2019; Mumuni & Mumuni, 2022; Wang et al., 2017). Much of the recent progress in data augmentation has been driven by advancements in black-box generative models, such as autoencoders (Kingma & Welling, 2022; Chadebec et al., 2022) and generative adversarial networks (GANs) (Antoniou, 2017).

**Interpretability.** Although there are data augmentation methods that do not rely on generative models (Maharana et al., 2022), these often depend on the knowledge of the underlying data generation mechanisms, which are typically unknown for complex datasets. As a result, creating interpretable augmented data involves interpreting black-box generative models, an area that remains an active research focus. To date, there is no fully satisfactory solution to this challenge. For example, the design of interpretable GANs is still evolving (Li et al., 2022b; She et al., 2021) and remains largely limited to specific domains, such as image generation.

### 2.2 DIMENSION REDUCTION AND RELATION TO DATA AUGMENTATION

**Linear Methods.** Classical *linear* dimension reduction techniques, such as Principal Component Analysis (PCA) (Wold et al., 1987) and Singular Value Decomposition (SVD) (Stewart, 1993), work by identifying the optimal linear subspace that minimizes reconstruction error, typically through the orthogonal projection of data onto this subspace. These methods are not only straightforward and explicit, but they also provide valuable geometric insights. For instance, PCA highlights the principal directions that capture the most variance in the data, uncovering important structural patterns.

However, one of the challenges in applying linear dimension reduction methods to data augmentation is the *inverse* problem, where reconstructing the original data from the space of reduced dimension is highly non-trivial. While some studies have explored indirect approaches to using linear dimension reduction for data augmentation (Abayomi-Alli et al., 2020; Sirakov et al., 2024), they are often application-specific and hard to generalize, limiting their broader applicability.

**Non-Linear Methods.** The *non-linear* generalizations, often called *manifold learning* (Meilă & Zhang, 2024), offer an alternative approach to dimension reduction. Popular methods like t-SNE (Hinton & Roweis, 2002; Van der Maaten & Hinton, 2008), Isomap (Tenenbaum et al., 2000), and UMAP (McInnes et al., 2018) are based on the manifold hypothesis, which suggests that high-dimensional data lie on a lower-dimensional manifold within the ambient space. The goal is to uncover this manifold and develop a smooth embedding that captures the data's intrinsic low-dimensional structure. While classical manifold learning methods do not rely on black-box neural networks, they are computationally complex, prone to overfitting, and require careful hyperparameter tuning, making interpretation challenging (Han et al., 2022).

In theory, manifold learning avoids the *inverse* problem by aiming to recover the underlying low-dimension manifold of the data with near-zero information loss, making it conceptually appealing for data augmentation. However, this is rarely achieved in practice, hence solving the *inverse* problem is still necessary to generate realistic augmented data. Additionally, classical manifold learning methods that do not rely on black-box neural networks are often limited to providing fixed embeddings for training data and cannot perform out-of-sample extensions (Duque et al., 2020), further limiting their ability to augment data. Recent approaches to address this limitation involve more complex algorithms (Coifman & Lafon, 2006; Williams & Seeger, 2000; Vladymyrov & Carreira-Perpinán, 2013) or the introduction of black-box generative models (Duque et al., 2020), which reintroduces the concern about interpretability.

## 3 PRELIMINARY

### 3.1 DUALLY-FLATNESS IN INFORMATION GEOMETRY

Information geometry studies the structure of *statistical manifolds* $\mathcal{S}$ within the space of probability distributions. In this paper, we are primarily concerned with the space of exponential families $\{p_\theta(x) \mid \theta \in \mathbb{R}^D\}$, where each $p_\theta$ denotes a probability density function parameterized by $\theta$. We focus on the key concept in this field, *dually-flatness*, in this preliminary, while directing readers to Amari (2016) for more comprehensive details.[1]

The starting point is the observation that the *log-partition function* $\psi(\theta)$ (also known as the *cumulant generating function* in statistics and *free energy* in physics) of an exponential family with density $p_\theta$ is convex in the *natural parameter* $\theta \in \mathbb{R}^D$. This convexity induces a natural coordinate system, $\theta$, on $\mathcal{S}$, defining both the Riemannian metric $g = \nabla^2 \psi(\theta)$ and the Bregman divergence (Bregman, 1967) $D_\psi(p_\theta, p_{\theta'})$. With these structures, the manifold $(\mathcal{S}, g)$ is flat, meaning that any curve $\theta(t) = at + b$ (where $a, b \in \mathbb{R}^D$ are constants) is a geodesic and lies entirely within $\mathcal{S}$. This flatness is known as *e-flatness*, and the geodesics are referred to as *e-geodesics* or *primal-geodesics*.

The dual structure arises from the *Legendre transform* (Legendre, 1787), which generates the dual function $\psi^*(\eta)$, where $\eta \in \mathbb{R}^D$ is the *expectation parameter*. This dual function is also convex, giving rise to the expectation coordinate system $\eta$, the dual Riemannian metric $g^*$, and also the dual Bregman divergence $D_{\psi^*}$ which is the well-known Kullback-Leibler divergence $D_{\mathrm{KL}}$. The corresponding flatness is termed *m-flatness*, with *m-geodesics* or *dual-geodesics* as its geodesics.

*Dually-flatness* then emerges from the interplay between these two structures. Specifically, for any point $p$ in $\mathcal{S}$, there is a unique point $p^*$ on an $e$-flat sub-manifold $\mathcal{B} \subseteq \mathcal{S}$ that minimizes the dual Bregman divergence $D_{\psi^*}(p, q) = D_{\mathrm{KL}}(p, q)$ (Amari, 2016, Theorem 1.5). This process, known as the *m-projection*, can be efficiently solved via convex optimization. The dual holds when switching $e$ and $m$ in this context. Projection is a central tool in information geometry with profound implications for understanding the geometry of $\mathcal{S}$, which we will utilize later.

### 3.2 STATISTICAL MANIFOLD ON POSETS

A set $\Omega$ is a *partially ordered set (poset)* if it is equipped with a *partial order* "$\leq$", a relation satisfying the following for all $x, y, z \in \Omega$: 1.) $x \leq x$ (reflexivity); 2.) $x \leq y$ and $y \leq x$ implies $x = y$ (antisymmetry); and 3.) $x \leq y$ and $y \leq z$ implies $x \leq z$ (transitivity). We focus on finite posets $\Omega$ with a bottom element $\perp$ such that $\perp \leq x$ for all $x \in \Omega$.

---

[1]We will assume some familiarity on the basic terminologies for manifold (Lee, 2012, Chapter 1, 4).

Given such a poset $\Omega$, consider a discrete random variable $X$ with finite support $\Omega$ with its probability mass function $p\colon \Omega \to \mathbb{R}_{\geq 0}$ being defined by $p(x) = \Pr(X = x)$ for $x \in \Omega$. A key observation is that for a discrete probability distribution $p$ over a poset $\Omega$, the *log-linear model on posets* recursively defines $\theta\colon \Omega \to \mathbb{R}$ as $\log p(x) =: \sum_{y \leq x} \theta(y)$ for all $x \in \Omega$. Intuitively, one can think of $\theta(x)$ for each $x \in \Omega$ as specifying the correct energy for $x$ that correctly represents $p(x)$, where the dependence between $\theta$'s on different elements depends on the poset structure. This model belongs to the exponential family, with $\theta$ corresponding to the natural parameters, except for $\theta(\bot)$ which coincides with the partition function. Thus, all discrete probability distributions over $\Omega$ form a $(|\Omega| - 1)$-dimensional dually-flat statistical manifold $\mathcal{S} := \{p\colon \Omega \to \mathbb{R}_{\geq 0} \mid \sum_{x \in \Omega} p(x) = 1\}$, with dual coordinate systems $(\theta, \eta)$ defined by the poset structure.

# 4 DATA AUGMENTATION WITH LOG-LINEAR MODEL ON POSETS

We first present our proposed framework in Section 4.1 and the backward projection algorithm in Section 4.2, then, we combine and apply them to data augmentation in Section 4.3. Finally, we discuss an important feature of the proposed method regarding interpretability in Section 4.4.

## 4.1 LOG-LINEAR MODEL ON POSETS FRAMEWORK

Given a dataset $\{z_i\}_{i=1}^n$, our proposed framework embeds the data into a statistical manifold $\mathcal{S}$ using an energy-based approach, leveraging the log-linear model on posets. This provides a geometric structure induced by the energy-based modeling, which is general and not restricted to any specific application, making it of broader interest. The process works in three steps: 1.) models each $z_i$ as a *real-valued poset*; 2.) embeds the data naturally into the statistical manifold $\mathcal{S}$; 3.) computes two coordinate representations of the embedded data using the log-linear model on posets. See Figure 1 for an illustration. We now explain each step in detail below.

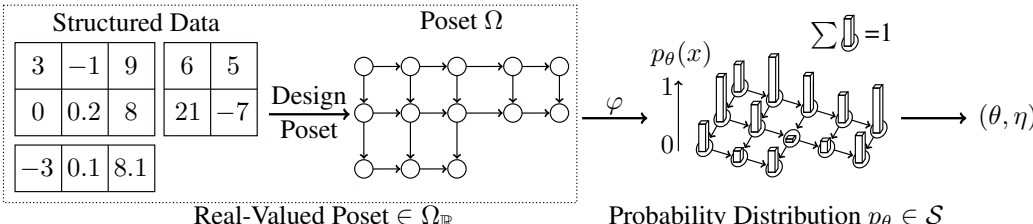

Figure 1: Given structured data, we design a corresponding poset $\Omega$ and embed the resulting real-valued poset as a discrete probability distribution $p_\theta(x)$ via a natural embedding $\varphi$ into the statistical manifold $\mathcal{S}$. Then the log-linear model on posets provides the dually-flat coordinates $(\theta, \eta)$ for $p_\theta$.

**Real-Valued Poset.** In the usual machine learning pipeline, inputs are constrained to be vectors or matrices, failing to deal with more complex data. In contrast, posets are flexible enough to capture data with structures, including vectors and matrices. For instance, focusing on the underlying data structure for now (i.e., omitting the feature associated with individual entry), a $D$-dimensional vector $z \in \mathbb{R}^D$ can be modeled by the poset $\Omega := [D]$ with the partial order being the natural order between positive integers. Similarly, other common data structures such as matrices or tensors can be treated in the same way, while capturing more complex structures potentially. In general, any data structure that naturally admits a partial order can be effectively modeled by a poset.

Now, considering the features associated with each entry in the data structure, we define the *real-valued poset*, which is a mapping from the poset $\Omega$ to, say, the set of real numbers $\mathbb{R}$ such that each entry (element) of the data structure (poset) $x \in \Omega$ is associated with a feature in $\mathbb{R}$. We denote the set of real-valued posets as $\Omega_\mathbb{R}$. In the $D$-dimensional vector example, $\Omega = [D]$, with each element $x \in \Omega$ corresponds to one of the $D$ dimensions. Associating a real number to each dimension (i.e., a $D$-dimensional vector) naturally corresponds to an element in $\Omega_\mathbb{R}$.

**Natural Embedding.** To embed the data $\{z_i \in \Omega_\mathbb{R}\}_{i=1}^n$, which are now modeled as real-valued posets, to the statistical manifold $\mathcal{S}$ which concerns with discrete probability distributions, we want

an embedding $\varphi \colon \Omega_{\mathbb{R}} \to \mathcal{S}$ such that $\sum_{x \in \Omega} (\varphi(z_i))_x = 1$ for all $z_i$ with $\dim(\mathcal{S}) = D - 1$.[2] From the perspective of energy-based modeling, $\varphi$ is oftentimes naturally induced, e.g., for tabular frequency data. Moreover, $\varphi$ often admits a natural inverse $\varphi^{-1}$, or an empirical one based on the data. We will take both $\varphi$ and $\varphi^{-1}$ as granted from now on.

**Dually-Flat Coordinates.** After defining $\varphi$, from the theory of information geometry and the log-linear model on posets, for each point $z_i' \coloneqq \varphi(z_i) \in \mathcal{S}$, we can associate the dually-flat coordinate systems $\theta(z_i') \in \mathbb{R}^{D-1}$ and $\eta(z_i') \in \mathbb{R}^{D-1}$. Such coordinate systems are with respect to the underlying poset structure of $\Omega$ and are driven by the principle of energy-based modeling.

## 4.2 Forward and backward projection

We now demonstrate how to incorporate projection theory to achieve data augmentation. As our algorithm is inspired from the architecture of autoencoders, we focus on two of the central building blocks: the *encoder* $\mathsf{Enc}(\cdot)$ and the *decoder* $\mathsf{Dec}(\cdot)$. First, for the encoding step, we formally explain how projection theory can be applied to perform dimension reduction within our framework. Next, for the decoding step, we introduce our proposed algorithm, termed *backward projection*, which serves as the *inverse* of dimension reduction. While our explanation is tailored to our proposed framework, i.e., the log-linear model on posets, the proposed backward projection algorithm itself is general and may be of independent interest as well.

**Dimension Reduction: Forward Projection.** Given the log-linear model on posets framework in Section 4.1, the embedding from $\Omega_{\mathbb{R}}$ to the statistical manifold $\mathcal{S}$ does not achieve dimension reduction as $\dim(\mathcal{S}) \approx \dim(\Omega_{\mathbb{R}})$. To achieve dimension reduction, we leverage the projection theory: by projecting $z_i' = \varphi(z_i)$ onto a low-dimensional flat *base sub-manifold* $\mathcal{B} \subseteq \mathcal{S}$ with $\dim(\mathcal{B}) \ll \dim(\mathcal{S})$, we obtain the desired *encoding* $\mathsf{Enc} \coloneqq \mathrm{Proj}_{\mathcal{B}} \circ \varphi \colon \Omega_{\mathbb{R}} \to \mathcal{B}$ that maps the data to a low-dimensional latent representation manifold. The encoding $\mathsf{Enc}(\cdot)$ is smooth and well-defined as the projection is unique when $\mathcal{B}$ is flat and minimizing either the primal or the dual Bregman divergence, depending on $\mathcal{B}$. These theoretical guarantees provide rigor and support reasoning through geometric intuition, which in turn offers interpretability.

**Inverse Dimension Reduction: Backward Projection.** As we have hinted at, one of the technical burdens is that the encoding $\mathsf{Enc}(\cdot)$ is not invertible, hence no natural decoding $\mathsf{Dec}(\cdot)$ is available, even when $\mathsf{Enc}(\cdot)$ only involves traditional linear dimension reduction algorithm. While finding the exact inverse is mathematically impossible as the pre-image of the projection is not unique in any sense (even in Euclidean space), here, we propose a simple, geometrically intuitive, and data-centric solution that aims to find the *inverse* of the projection that is similar to the original data.

The high-level intuition is simple: if the result of the projection is close, then so is the original data, i.e., its *inverse*. Hence, given a point in the low-dimensional latent representation space, we try to "project it back" to approximate the original dataset by exploiting the fact that we have access to the *inverse* of the dataset's projection, i.e., the dataset itself. Specifically, we can artificially create a *local* sub-manifold around a subset of the dataset, determined by the nearest neighbors of that given point in the latent representation space, and *backward* project onto it.

Formally, assuming that we have access to the embedded dataset $\{z_i' = \varphi(z_i)\}_{i=1}^n$ and their projected result $\{w_i = \mathrm{Proj}_{\mathcal{B}}(z_i')\}_{i=1}^n$ for some base sub-manifold $\mathcal{B}$. To find the *inverse* of some given point $w^* \in \mathcal{B}$ assuming it comes from the projection on $\mathcal{B}$, we first find $w^*$'s $k$-nearest neighbors among $w_i$'s, obtaining a size $k$ index set $N$. Then we create a *local data sub-manifold* $\mathcal{D}$ based on the pre-images $z_i'$'s of these $w_i$'s, and project $w^*$ on $\mathcal{D}$ to obtain the *inverse* $z'^* = \mathrm{Proj}_{\mathcal{B}}^{-1}(w^*) \coloneqq \mathrm{Proj}_{\mathcal{D}}(w^*)$. Algorithm 4.1 summarizes this procedure, which we termed *backward projection*. With access to $\mathrm{Proj}_{\mathcal{B}}^{-1}(\cdot)$, decoding is simply $\mathsf{Dec} \coloneqq \varphi^{-1} \circ \mathrm{Proj}_{\mathcal{B}}^{-1} \colon \mathcal{B} \to \Omega_{\mathbb{R}}$, serving as the *inverse* of $\mathsf{Enc}(\cdot)$ as desired.

**Remark 4.1.** *A flat sub-manifold can be defined by forcing linear constraints on the ($\theta$ or $\eta$) coordinates. For instance, given the nearest neighbors $z_{i\star}'$, one can define $\mathcal{D} \coloneqq \{\theta \in \mathbb{R}^{\dim(\mathcal{S})} \mid (\theta)_x = (\theta(z_{i\star}'))_x\}$ for some $x \in \Omega$, namely, we fix some indexes to be the corresponding $\theta$-coordinate values of $z_{i\star}'$. The quality of Algorithm 4.1 can be controlled by choosing appropriate linear constraints.*

---

[2]In fact, we can also consider the manifold of positive measures, which avoids the dimension being $D - 1$ and the potential scaling issues. We omit this trivial extension in the presentation to prevent complications.

---

**Algorithm 4.1:** Backward Projection

---

**Data:** A data point $w^* \in \mathcal{B}$, $\varphi$-embedded dataset $\{z_i'\}_{i=1}^n$, projection result $\{w_i\}_{i=}^n$ on $\mathcal{B}$, $k \in \mathbb{N}$
**Result:** Backward projected data $z'^*$

1   $N \leftarrow \texttt{Nearest-Neighbor}(k, w^*, \{w_i\}_{i=1}^n)$        `// N ⊆ [n] with |N| = k`
2   $\mathcal{D} \leftarrow \texttt{Sub-Manifold}(\{z_i'\}_{i \in N})$
3   $z'^* \leftarrow \texttt{Projection}(w^*, \mathcal{D})$
4   **return** $z'^*$

---

Algorithm 4.1 is a geometrically intuitive, data-centric algorithm with desirable theoretical guarantees such as divergence minimizing when projecting on the constructed local data sub-manifold $\mathcal{D}$. Its white-box nature ensures a level of interpretability, making it the cornerstone of our method, in contrast to black-box generative models.

### 4.3 DATA AUGMENTATION WITH LOG-LINEAR MODEL ON POSETS

With all the building blocks in place, we can now formally describe the complete data augmentation algorithm, which consists of three phases: 1.) encoding, 2.) generating, and 3.) decoding.

**Encoding.** As described in Section 4.2, the encoding $\mathsf{Enc} := \mathrm{Proj}_{\mathcal{B}} \circ \varphi$ is simply a combination of the natural embedding followed by a projection. Notation-wise, we write $w_i := \mathsf{Enc}(z_i)$.

**Generating.** To generate new data $z^*$, we first generate a new point $w^*$ in the latent space, which in our case, is a pre-specified flat base sub-manifold $\mathcal{B}$. This can be done in various ways, such as using pure heuristics, controlled perturbations, or even black-box generative models. In our case, we focus on a simple, white-box generation method: *kernel density estimation* (Davis et al., 2011; Parzen, 1962). Specifically, we first fit a kernel density estimation model $M$ on either the $\theta$ or $\eta$ coordinate systems, then sample $m$ new points $w^*$ in the latent space $\mathcal{B}$ from $M$.

**Decoding.** As described in Section 4.2, the decoding $\mathsf{Dec} := \varphi^{-1} \circ \mathrm{Proj}_{\mathcal{B}}^{-1}$ is simply a combination of backward projection (Algorithm 4.1) with the inverse of the natural embedding. Notation-wise, we write $z^* := \mathsf{Dec}(w^*) = \varphi^{-1}(z'^*)$ where $z'^* := \mathrm{Proj}_{\mathcal{B}}^{-1}(w^*) := \mathrm{Proj}_{\mathcal{D}}(w^*)$.

We summarize the above procedure in Algorithm 4.2.

---

**Algorithm 4.2:** Data Augmentation with Log-Linear Model on Posets

---

**Data:** A dataset $\{z_i\}_{i=1}^n$, embedding $\varphi \colon \Omega_{\mathbb{R}} \to \mathcal{S}$, $k \in \mathbb{N}$, flat base sub-manifold $\mathcal{B}$, size $m \in \mathbb{N}$
**Result:** A generated dataset $\{z_j^*\}_{j=1}^m$ of size $m$

1   **for** $i = 1, \ldots, n$ **do**                                  `// Encoding`
2      $z_i' \leftarrow \varphi(z_i)$
3      $w_i \leftarrow \texttt{Projection}(z_i', \mathcal{B})$             `// w = Enc(z) = Proj_B ∘φ(z)`
4
5   $\{w_i^*\}_{i=1}^m \leftarrow \texttt{Sample}(\{w_i\}_{i=1}^n, \mathcal{B}, m)$       `// Generating m points`
6
7   **for** $j = 1, \ldots, m$ **do**                                `// Decoding`
8      $z_j'^* \leftarrow \texttt{Backward-Projection}(w_j^*, \{z_i'\}_{i=1}^n, \{w_i\}_{i=1}^n, k)$    `// Algorithm 4.1`
9      $z_j^* \leftarrow \varphi^{-1}(z_j'^*)$              `// z* = Dec(w*) = φ^{-1} ∘ Proj_B^{-1}(w*)`
10   **return** $\{z_j^*\}_{j=1}^m$

---

In what follows, we use *positive tensors* as the running example for a better illustration.

**Example 4.2** (Positive tensor). *A $d^{th}$-order tensor $T \in \mathbb{R}^{I_1 \times \cdots \times I_d} =: \mathbb{R}^D$ is a multidimensional array with real entries for every index vector $v = (i_1, \ldots, i_d) \in [I_1] \times \cdots \times [I_d] =: \Omega$ where for each $k$, $[I_k] := \{1, 2, \ldots, I_k\}$ for a positive integer $I_k$. Tensors with entries all being positive are called positive tensors, denoted as $P \in \mathbb{R}_{\geq 0}^{I_1 \times \cdots \times I_d}$. For tensors, a natural partial order "$\leq$" one can impose on $\Omega$ between two index vectors $v = (i_1, \ldots, i_d)$, $w = (j_1, \ldots, j_d)$ is that $v \leq w$ if and only if $i_k \leq j_k$ for all $k = 1, \ldots, d$. Finally, for positive tensors, a simple embedding $\varphi \colon \mathbb{R}_{\geq 0}^{I_1 \times \cdots \times I_d} \to \mathcal{S}$ where $P' := \varphi(P) \colon \Omega \to \mathbb{R}_{\geq 0}$ such that $P_v' := P_v / \sum_{w \in \Omega} P_w$ for all $v \in \Omega$ can be defined.*

We now illustrate Algorithm 4.2 with positive tensors. Following the notations in Example 4.2, let's write $z_i$'s as $P_i$'s, $z_i'$'s as $P_i'$'s, and $w_i$'s as $Q_i$'s. Firstly, Example 4.2 provides one way to model positive tensors by real-valued posets and define a natural embedding $\varphi$ (i.e., normalization), giving $P_i'$. To obtain the final encoding, we choose some base manifold $\mathcal{B}$ to project $P_i'$ onto, giving $Q_i := \mathrm{Proj}_{\mathcal{B}}(P_i')$. For generation, we simply fit a kernel density estimation model $M$ to $\{w_i = Q_i\}_{i=1}^n$ and sample a new $Q^* \sim M$. Finally, for the decoding step, consider the case of $k = 1$, $\mathcal{D}$ is created by some linear constraints w.r.t. one particular $P_{i^\star}'$,

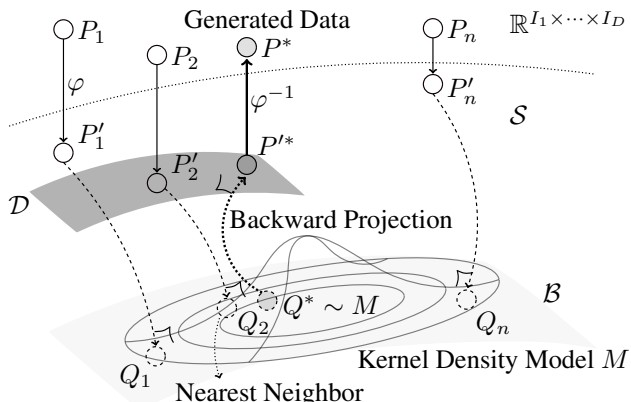

Figure 2: Data augmentation for positive tensors via Algorithm 4.2 with $k = 1$ and $\mathcal{D} = \mathcal{D}_{P_2'}$.

where $i^\star$ corresponds to the nearest neighbor $Q_{i^\star}$ of $Q^*$ among $Q_i$'s. We then backward project $Q^*$ on $\mathcal{D}$ with Algorithm 4.1 to obtain $P'^* := \mathrm{Proj}_{\mathcal{D}}(Q^*)$, and we output $P^* := \varphi^{-1}(P'^*)$ as our generated result.[3] See Figure 2 for an illustration, where we let $k = m = 1$.

Algorithm 4.2 integrates both forward (encoding) and backward (decoding) projections, which, as discussed in Section 4.2, are interpretable due to its white-box nature and come with strong theoretical guarantees. When the generating step is performed in a clear and white-box manner, Algorithm 4.2 retains its interpretability while continuing to benefit from these theoretical guarantees.

### 4.4 CONSTRUCTION OF SUB-MANIFOLDS

For any sub-manifold $\mathcal{S}' \subseteq \mathcal{S}$, as $\dim(\mathcal{S}')$ increases, more information of the data is preserved after forward projection onto $\mathcal{S}'$. In the case of constructing the base sub-manifold $\mathcal{B}$, the quality of the backward projection $\mathrm{Proj}_{\mathcal{B}}^{-1}(\cdot)$ (Algorithm 4.1) should increase along with $\dim(\mathcal{B})$ for the same reason. However, in the extreme case where $\dim(\mathcal{B}) \approx \dim(\mathcal{S})$, Algorithm 4.2 becomes less effective due to the sparsity of the data, resulting in an intrinsic trade-off for choosing $\dim(\mathcal{B})$ (see Appendix A.3 for an empirical justification). In this section, we argue that by leveraging existing tools and understandings of the log-linear model on posets, such an intrinsic trade-off for constructing sub-manifolds (either $\mathcal{B}$ or $\mathcal{D}$) provides an additional layer of interpretability and control compared to black-box generative models like autoencoders.

To keep our presentation concise and concrete, we focus on positive tensors, although the argument and the high-level idea extend to more general cases. Firstly, the projection theory is well-explored for positive tensors within the log-linear model, where several established constructions for flat base sub-manifolds $\mathcal{B} \subseteq \mathcal{S}$ (Sugiyama et al., 2018; Ghalamkari et al., 2024) have proven powerful in capturing the non-trivial structure of positive tensors after the projection. One of which is called the *many-body tensor approximation* (Ghalamkari et al., 2024), which captures a *hierarchy* of mode interactions with different $\dim(\mathcal{B})$. Specifically, the $\ell$-body approximation considers projection on the following sub-manifold

$$\mathcal{M}_\ell := \{\theta \in \mathbb{R}^{\dim(\mathcal{S})} \mid (\theta)_x = 0 \text{ for all } \textbf{non } \ell\text{-body parameters } x \in \Omega\}, \qquad (1)$$

where the $\ell$-body parameter corresponds to $\ell$ non-one indices, acting as a generalization of one-body and two-body parameters (Ghalamkari & Sugiyama, 2022). Intuitively speaking, an $\ell$-body parameter captures the interaction among $\ell$ different modes, hence, when $\mathcal{B} = \mathcal{M}_\ell$, all interactions between modes of orders higher than $\ell$ are neglected. This offers a practical design choice for employing Algorithm 4.2. In particular, it allows us to leverage prior knowledge of the data to design an appropriate base sub-manifold $\mathcal{B}$ and also the local data sub-manifold $\mathcal{D}$ that captures different degrees of information with appropriate dimension. This approach provides a more principled way of defining the latent space, compared to black-box models like autoencoders, where the latent space dimensions are oftentimes tuned without a clear understanding of what those dimensions represent.

---

[3]Empirically, we let $\varphi^{-1}$ to be *reversing the average of original scaling among the nearest neighbors.*

## 5 EXPERIMENT

In this section, we conduct a series of experiments to validate the efficacy of our proposed data augmentation method. We focus primarily on image tasks for a clear illustration, where we compare our method with autoencoder models. Additional experiments can be found in Appendix A.

### 5.1 SETUP

Here, we briefly summarize the experimental setup, while directing readers to Appendix A.1 for more details. Consider the image classification task[4] on the MNIST dataset (LeCun, 1998), with the training set size being $1000$ ($200$ samples for each digit). Since MNIST images are in $\mathbb{R}_{\geq 0}^{28\times 28}$, we apply the log-linear model with posets for positive tensors as in Example 4.2. For the sub-manifold constructions, we utilize the many-body approximation (Equation (1)) with its variances when constructing the base sub-manifold $\mathcal{B}$ and the local data sub-manifold $\mathcal{D}$, where we use $\dim(\mathcal{B}) = 17$ and $\dim(\mathcal{D}) = 767$. For a fair comparison with the autoencoder model, we consider a simple $2 + 2$ layers architecture with latent space dimension $17 = \dim(\mathcal{B})$. Finally, when generating data, we first fit a kernel density estimation model $M$ on the latent representation of the training dataset, sample a new latent representation from which, and then decode it. We note that important hyperparameters ($k = 8$ and the bandwidth $0.01$ of $M$) are chosen via a simple grid search in Appendix A.2.

### 5.2 VISUAL INSPECTION

To illustrate how Algorithm 4.2 works in practice, Figure 3 shows the intermediate results after projection onto $\mathcal{B}$, while Figure 4 shows the results of Algorithm 4.2 after applying backward projection (Algorithm 4.1). We emphasize that the results from Figure 4 do not come from backward projecting the results of Figure 3; instead, they come from the latent representations sampled from $M$.

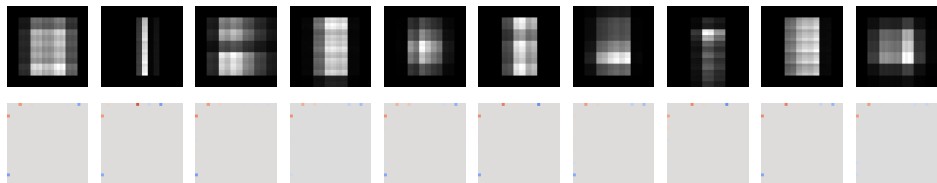

Figure 3: (*Top*) Forward projected data on $\mathcal{B}$. (*Bottom*) Heat map of corresponding $\theta$ values.

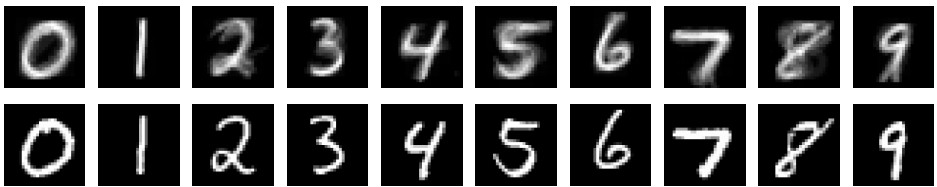

Figure 4: (*Top*) Augmented data via Algorithm 4.2. (*Bottom*) The closest training data.

For comparison, Figure 5 shows the data augmentation results generated by the autoencoder. Despite careful bandwidth tuning when fitting the kernel density model, the autoencoder results appear to overfit the training set. Finally, we note an interesting difference between the two approaches: our proposed method produces a *blurred* effect, while the autoencoder exhibits *hard-clipping*.

### 5.3 CLASSIFICATION PERFORMANCE

We evaluate our proposed method on the downstream task, i.e., classification performance, in addition to visual inspection. Specifically, we train a linear classifier on three types of training datasets: 1.) original dataset, 2.) augmented dataset, and 3.) original dataset combined with the augmented

---

[4]As there are only finitely many labels (classes) in classification tasks, one can perform Algorithm 4.2 for each class separately without worrying about assigning labels.

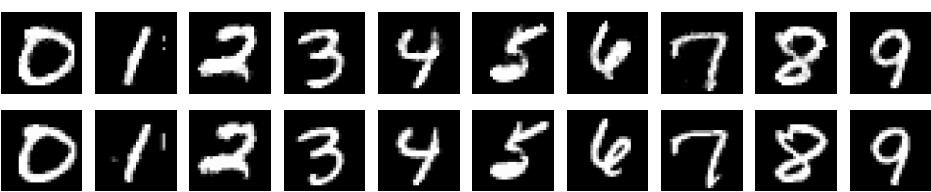

Figure 5: (*Top*) Augmented data via autoencoder. (*Bottom*) The closest training data.

dataset, where the augmented dataset consists of $m = 200$ augmented samples, which is 20% of the original training set. For clarity, we refer to the original dataset as **Original**, the dataset augmented with Algorithm 4.2 as **Ours**, and the dataset augmented using the autoencoder model as **AE**.

The results are shown in Table 1, where each test set consists of 500 samples, evaluated over 20 bootstrapping runs. Firstly, observe that **Original**+**Ours** and **Original**+**AE** outperforms **Original** as expected, with the former outperforming the latter slightly. Moreover, we see that **Ours** outperforms **AE** by a large margin, which is surprising given the representation power of the autoencoder compared to our fully white-box, interpretable method. Overall, our method achieves competitive performance against black-box generative models in the downstream task while offering interpretability. We direct readers to Appendix A.5 for additional evaluations on other datasets.

Table 1: Test accuracy of the linear classifier trained on different training sets.

| Training Set | Original | Ours | AE | Original+Ours | Original+AE |
|---|---|---|---|---|---|
| **Accuracy** | $81.79 \pm 4.57\%$ | $75.37 \pm 2.89\%$ | $68.12 \pm 3.96\%$ | $83.40 \pm 3.22\%$ | $82.72 \pm 3.50\%$ |

## 5.4 INTERPRETABILITY WITH CHOICES OF SUB-MANIFOLDS

As discussed in Section 4.4, constructing the base sub-manifold carefully allows for an additional layer of interpretability and control. In Section 5.1, the default base manifold $\mathcal{B}$, though implicit, is $\mathcal{M}_1$ for the tensor structure $\mathbb{R}_{\geq 0}^{7 \times 2 \times 2 \times 7 \times 2 \times 2}$. We now consider $\mathcal{B} = \mathcal{M}_\ell$ for $\ell = 1$ to $3$ for comparison, while direct readers to Appendix A.4 for a more in-depth experiment. Following the same setup as in Figures 3 and 4 for $\ell = 1$, the results for $\ell = 2, 3$ are shown in Figures 6 and 7.

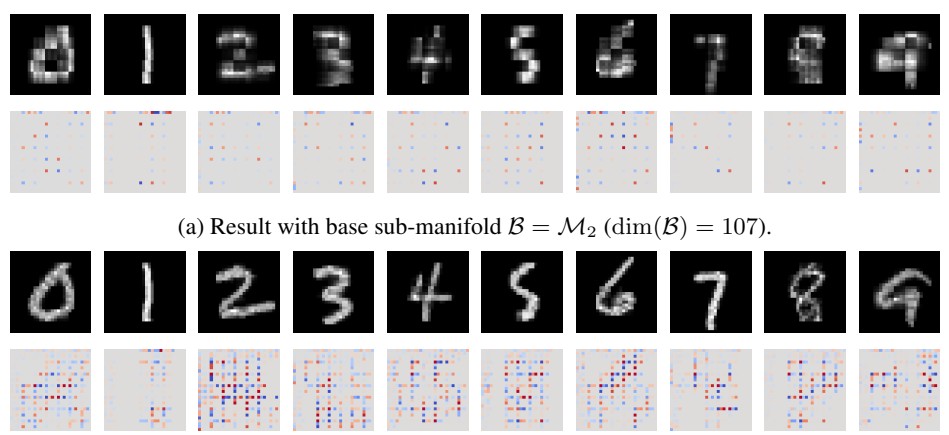

(a) Result with base sub-manifold $\mathcal{B} = \mathcal{M}_2$ ($\dim(\mathcal{B}) = 107$).

(b) Result with base sub-manifold $\mathcal{B} = \mathcal{M}_3$ ($\dim(\mathcal{B}) = 327$).

Figure 6: (*Top*) Forward projected data. (*Bottom*) Heat map of corresponding $\theta$ values.

We observe several interesting phenomena. First, for $\ell = 1$ with a small base sub-manifold dimension, the forward projection results (Figure 3) appear visually unclear, in contrast to the augmentation results (Figure 4). Note that throughout the experiment, the local data sub-manifold $\mathcal{D}$ has a dimension of 767, indicating a high degree of freedom for backward projection. This suggests that

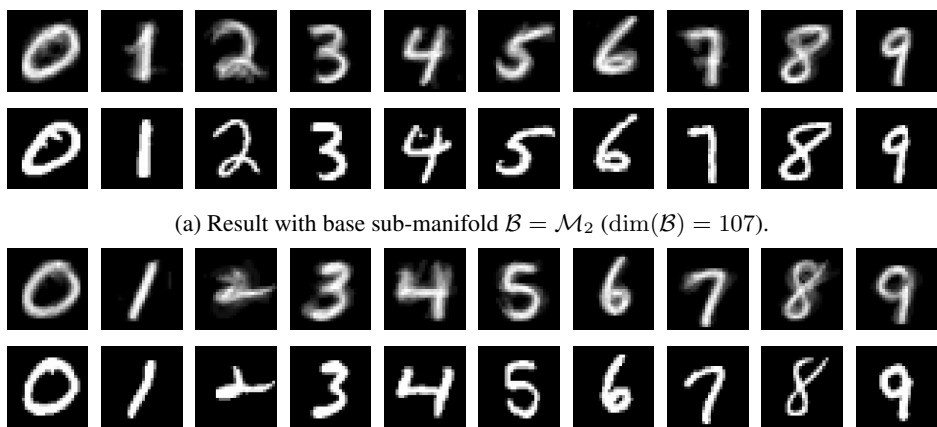

(a) Result with base sub-manifold $\mathcal{B} = \mathcal{M}_2$ ($\dim(\mathcal{B}) = 107$).

(b) Result with base sub-manifold $\mathcal{B} = \mathcal{M}_3$ ($\dim(\mathcal{B}) = 327$).

Figure 7: (*Top*) Augmented data via Algorithm 4.2. (*Bottom*) The closest training data.

$\mathcal{M}_1$ can effectively capture key features across signals, leading to non-trivial neighbor information and thus constructing a sufficiently good local data sub-manifold. Second, as expected, the higher the dimension of $\mathcal{M}_\ell$ (i.e., value of $\ell$), the more signal structures (in terms of mode interactions) are preserved as shown in Figures 3 and 6, resulting in better performance (Figures 4 and 7).

Based on the theory of many-body approximation, one can construct the base sub-manifold with a clear understanding of the trade-off between dimensionality and the performance of Algorithm 4.2. Unlike black-box generative models, which often rely on heuristics or blindly tuning the latent space dimension, our proposed method offers an additional layer of interpretability.

## 6 DISCUSSION

**Structural Limitation.** While the log-linear model is flexible to represent structural data, it still faces limitations. The key issue lies in the model's reliance on a partial order of the index set, which makes it impossible to ensure invariance under the permutation of indexes. For instance, modeling graphical data usually requires non-invariance and non-equivariance of vertices (i.e., indexes), in this case, the log-linear model might not be the best model due to its structural limitations.

**Meta-Perspective.** Classical information geometry typically involves learning a single distribution by manipulating a single point in the statistical manifold $\mathcal{S}$, as seen in tasks like learning the Boltzmann machine or finding the maximum likelihood estimation (Amari, 2016). In our case, however, we treat data as probability distributions within $\mathcal{S}$, offering a new perspective for applying the information geometry framework. With multiple distributions in $\mathcal{S}$, a natural extension would be to employ data-centric machine learning algorithms to learn the "data" distribution, i.e., the distribution of these *distributions*, thereby providing a *meta*-perspective.

## 7 CONCLUSION

In this paper, we proposed a novel data augmentation method that leverages several information geometric algorithms, incorporating interpretability while maintaining competitive performance. Our framework, built on the log-linear model on posets, equips data with information geometric structures, facilitating geometric reasoning and algorithm design. The proposed backward projection algorithm reverses the dimension reduction process in a geometrically intuitive and data-centric manner, which may be of independent interest.

We empirically demonstrated that our method achieves competitive performance compared to traditional autoencoder-based approaches on downstream tasks, even though the latter may possess greater representational power but lack interpretability, which is a crucial requirement in many practical applications. Overall, our work paves the way for further exploration of information geometric algorithms in various domains, not limited to data augmentation.

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

## A  OMITTED DETAILS FROM SECTION 5

### A.1  DETAILS OF IMAGE CLASSIFICATION SETUP

**Log-Linear Model on MNIST.**  When we apply the log-linear model on posets for positive tensors, we first reshape every image into $\mathbb{R}_{\geq 0}^{7\times 2\times 2\times 7\times 2\times 2}$ and consider the natural poset $\Omega$ corresponding to this $6^{\text{th}}$-order tensor structure. From the discussion in Section 4.4, when considering the image as a tensor of shape $(7, 2, 2, 7, 2, 2)$ instead of $(28, 28)$, a finer hierarchy of projection is possible via many-body approximation (Ghalamkari et al., 2024). In Section 5, the default choice of the base sub-manifold is $\mathcal{B} = \mathcal{M}_1$ for the $6^{\text{th}}$-order tensor structure $\mathbb{R}_{\geq 0}^{7\times 2\times 2\times 7\times 2\times 2}$ as defined in Equation (1). On the other hand, the local sub-manifold $\mathcal{D}$ is constructed by fixing every one-body parameter: given a set $N$ of $k$ nearest neighbors, we let $\mathcal{D}$ as $\mathcal{M}_1^*(N)$ where in general, we define

$$\mathcal{M}_\ell^*(N) := \left\{ \theta \in \mathbb{R}^{\dim(\mathcal{S})} \mid (\theta)_x = \frac{1}{k} \sum_{i^* \in N} \left( \theta(z'_{i^*}) \right)_x \text{ for all } \ell\text{-body parameters } x \in \Omega \right\}, \quad (2)$$

where $\dim(\mathcal{S}) = 28 \times 28 = 784$ for MNIST. This is like the dual notion of $\mathcal{M}_\ell$: in Equation (1), we allow all $\ell$-body parameters to vary; here, we allow all **non** $\ell$-body parameters to vary.

**Kernel Density Estimation Model.**  The default bandwidth for the kernel density estimation model is set to be $0.05$ to avoid overfitting. This is a fair comparison since the latent space dimensions for our proposed method and the autoencoder model we consider are the same.

**Linear Classifier.**  The classification task is conducted with a simple linear classifier trained with Stochastic Gradient Descent (SGD) (Ruder, 2016) till convergence with a learning rate of $0.01$.

### A.2  SENSITIVITY AND ROBUSTNESS

We examine our proposed method's robustness and sensitivity of the *bandwidth* used when fitting the kernel density model, and also the *number $k$ of the nearest neighbors* used in Algorithm 4.1.

**Bandwidth of Kernel Density Estimation Model.**  Consider varying the bandwidth we use when fitting the kernel density model, ranging among $\{0.01, 0.05, 0.1, 0.2, 0.5\}$. The results are shown in Figure 8, where we omit showing the closest training data as it is not important for the purpose here. We observe that Algorithm 4.2 is robust under different bandwidths when working with the kernel density estimation model in the generating step.

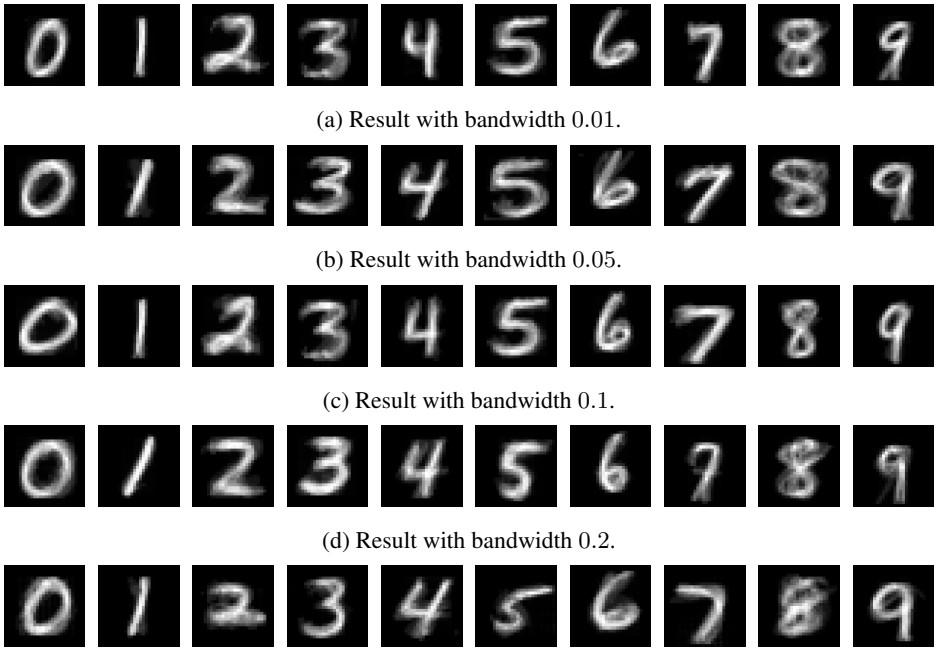

(a) Result with bandwidth 0.01.

(b) Result with bandwidth 0.05.

(c) Result with bandwidth 0.1.

(d) Result with bandwidth 0.2.

(e) Result with bandwidth 0.5.

Figure 8: Augmented data via Algorithm 4.2 with different kernel density estimation bandwidths.

**Number of Nearest Neighbors.** Next, we consider ranging $k$ among $\{1, 4, 8, 16\}$. The results are shown in Figure 9. Observe that when $k$ is small, e.g., 1, the result of Algorithm 4.2 tends to overfit since the local sub-manifold $\mathcal{D}$ in Algorithm 4.1 is defined using only the nearest neighbor. When $k$ goes up, a non-trivial augmentation emerges, robust across different $k$'s.

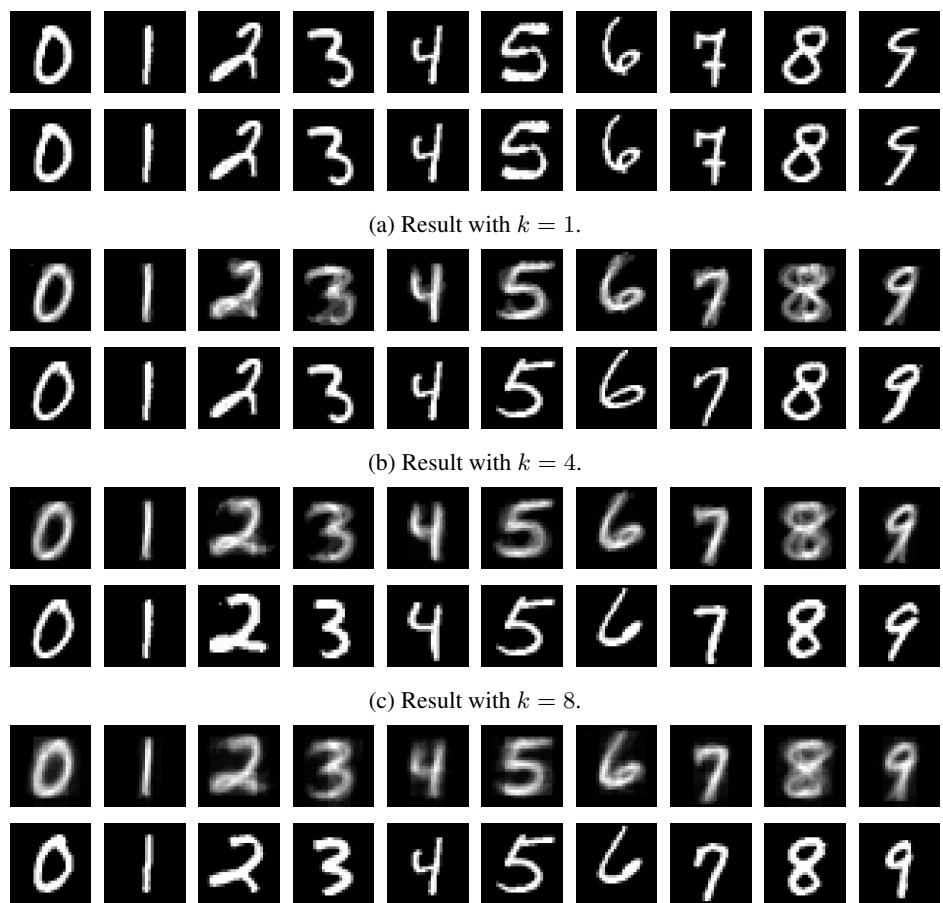

(a) Result with $k = 1$.

(b) Result with $k = 4$.

(c) Result with $k = 8$.

(d) Result with $k = 16$.

Figure 9: (*Top*) Augmented data via Algorithm 4.2 with different $k$'s for Algorithm 4.1. (*Bottom*) The closest training data.

### A.3 NECESSITY OF DIMENSION REDUCTION

We demonstrate that dimension reduction, a key building block of our proposed method based on the intuition we have from autoencoder-like models, is necessary for Algorithm 4.2 to work.

**Direct Fitting.** As discussed, naive perturbation-based data augmentation methods fall short of high-dimensional data due to the sparsity of the data. Figure 10 shows the results of directly fitting a kernel density estimation model on MNIST with 1000 samples.

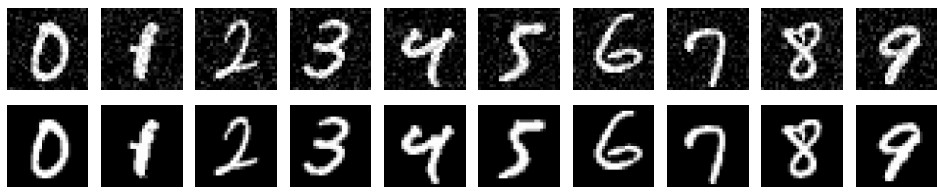

Figure 10: (*Top*) Augmented data via directly fitting a kernel density estimation model with a bandwidth 30. (*Bottom*) The closest training data.

Observe that even with a large bandwidth (30) to introduce variability, we only see meaningless noisy perturbation on one of the exact training samples, indicating overfitting.

**Local Data Sub-Manifold.** A potential problem related to the necessity of dimension reduction is that, if $\mathcal{D}$ captures too much local information about the data, backward projecting a random latent representation $w^* \in \mathcal{B}$ might already suffice to augment the data in a non-trivial way, without the need for knowing the latent representations of the training dataset. To this end, consider sampling uniformly random latent representations within the empirical range we observed from the latent representations of the training data and perform Algorithm 4.1. The results are shown in Figure 11.

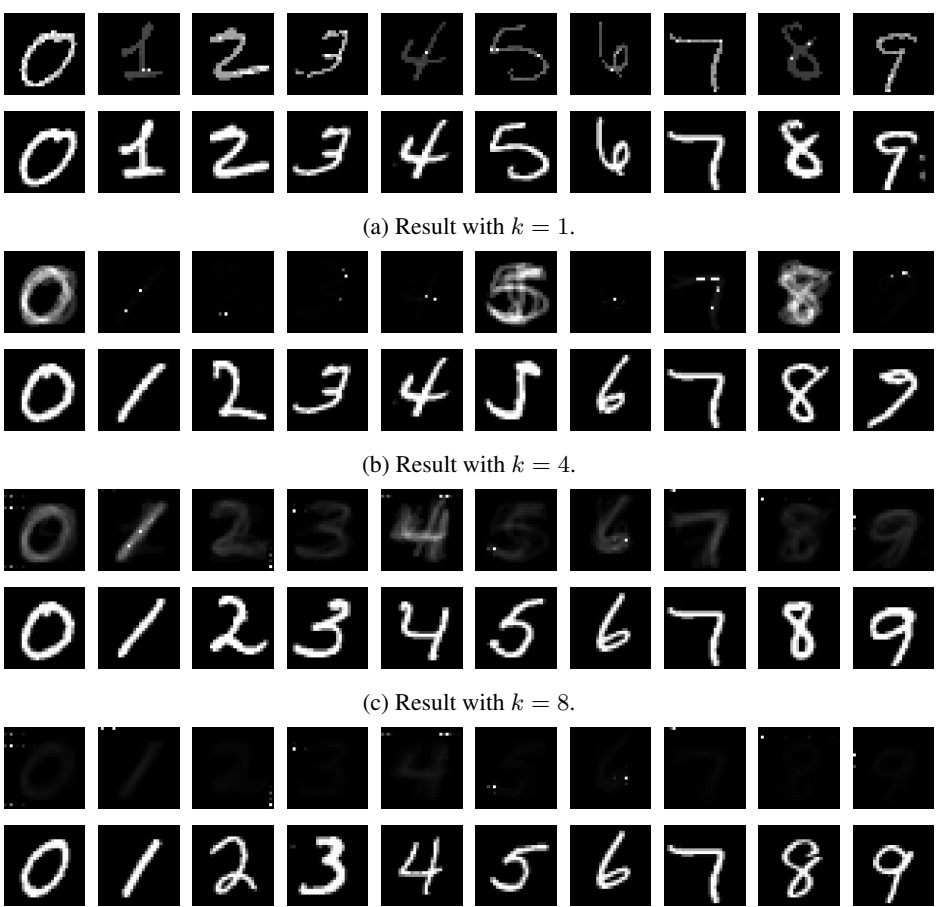

(a) Result with $k = 1$.

(b) Result with $k = 4$.

(c) Result with $k = 8$.

(d) Result with $k = 16$.

Figure 11: (*Top*) Augmented data on random latent representations via Algorithm 4.2 with different $k$'s for Algorithm 4.1. (*Bottom*) The closest training data.

For $k = 1$, Figure 11(a) shows that similar to Figure 9(a), it is possible to overfit one of the training data (i.e., the nearest neighbor of the randomly sampled latent representation). This is not surprising since the base sub-manifold is only of dimension 17 and the local data sub-manifold is of dimension 767, as the random latent representation is sufficiently close to one of the representations of the training data in $\mathcal{B}$, their backward projection result should not deviate too much. Furthermore, we observe the *fading effect*, which intuitively corresponds to *misspecification of the energy*, indicating that the sampled latent representation is fundamentally different from the one of the dataset.

As $k$ increases, the reason for getting informative and meaningful latent representations from the original dataset becomes clear. Specifically, we start to see *degeneration*: from unclear overlappings to collapsing (i.e., only a few pixels are showing). Intuitively speaking, it is because the random latent representation's nearest neighbors appear to be significantly different, hence failing to provide a consistent local data sub-manifold. For instance, in the extreme case when $k = 16$, the local data sub-manifold is completely not informative, resulting in collapsing. Overall, without dimension

reduction, we will lose the reference of *realisitc latent representations* provided by the original dataset, which leads to bad performance once we are beyond the trivial overfitting regime.

## A.4 CHOICES OF TENSOR STRUCTURE AND CONSTRUCTION OF SUB-MANIFOLDS

In Section 5.4, we consider varying $\ell$ for $\mathcal{B} = \mathcal{M}_\ell$ with the tensor structure being $\mathbb{R}_{\geq 0}^{7 \times 2 \times 2 \times 7 \times 2 \times 2}$. In this section, we further vary the tensor structure as well: in particular, we consider the tensor structure of the MNIST image being $\mathbb{R}_{\geq 0}^{28 \times 28}$, $\mathbb{R}_{\geq 0}^{7 \times 4 \times 7 \times 4}$, and $\mathbb{R}_{\geq 0}^{7 \times 2 \times 2 \times 7 \times 2 \times 2}$. For notation convenience, we write their corresponding poset structures as $\Omega_{28 \times 28}$, $\Omega_{7 \times 4 \times 7 \times 4}$, and $\Omega_{7 \times 2 \times 2 \times 7 \times 2 \times 2}$, and further write the many-body approximation sub-manifold (Equation (1)) as $\mathcal{M}_\ell(\Omega)$ and its dual (Equation (2)) as $\mathcal{M}_\ell^*(N, \Omega)$ for a particular poset $\Omega$ to emphasize the dependency. In particular, throughout this section, we consider the default local data sub-manifold, i.e., $\mathcal{D} = \mathcal{M}_\ell^*(N, \Omega_{7 \times 2 \times 2 \times 7 \times 2 \times 2})$, for consistency. Finally, we consider ranging $\ell$ from 1 to at most 4 where we neglect the degenerate case: for instance, in the case of $\Omega_{28 \times 28}$, $\mathcal{M}_2(\Omega_{28 \times 28}) = \mathcal{S}$ as there are only two modes for a matrix, therefore degenerates to direct fitting which is not of interest (see Appendix A.3).

The results for the finest structure $\mathbb{R}_{\geq 0}^{7 \times 2 \times 2 \times 7 \times 2 \times 2}$ are shown in Figures 12 and 13. As $\ell$ grows, the forward projection results in Figure 12 preserve the structure of the data better, subsequently improving the quality of the augmented data, as shown in Figure 13.

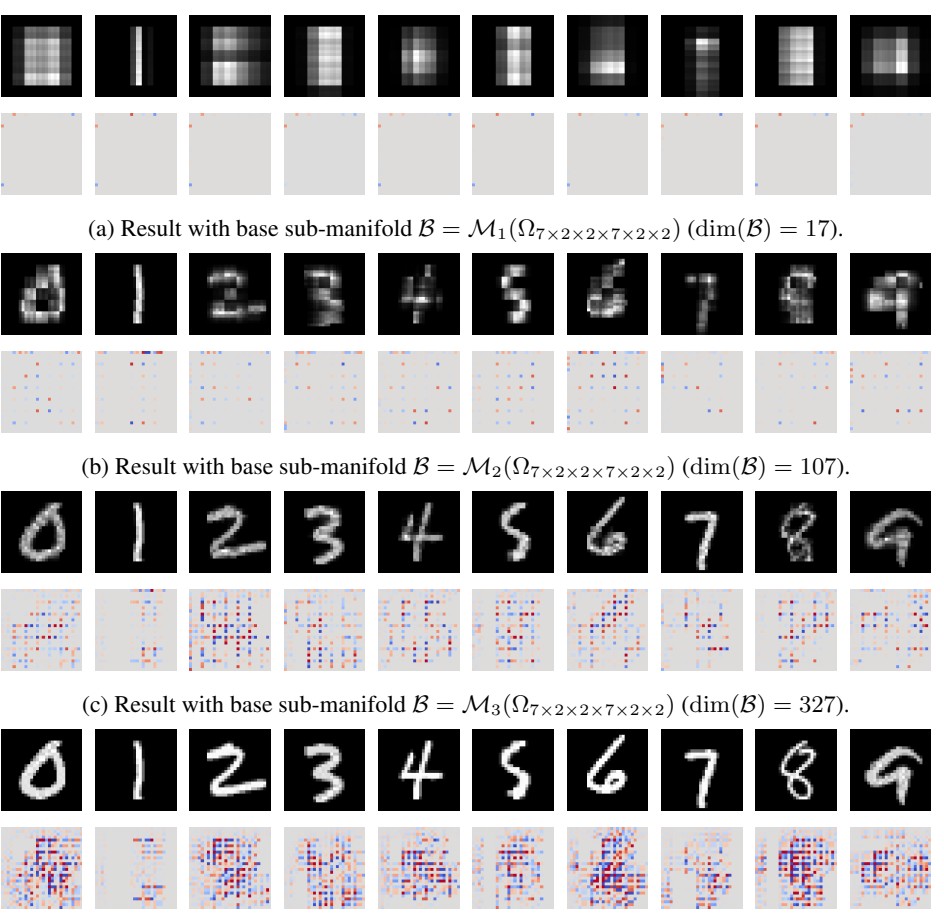

(a) Result with base sub-manifold $\mathcal{B} = \mathcal{M}_1(\Omega_{7 \times 2 \times 2 \times 7 \times 2 \times 2})$ (dim($\mathcal{B}$) = 17).

(b) Result with base sub-manifold $\mathcal{B} = \mathcal{M}_2(\Omega_{7 \times 2 \times 2 \times 7 \times 2 \times 2})$ (dim($\mathcal{B}$) = 107).

(c) Result with base sub-manifold $\mathcal{B} = \mathcal{M}_3(\Omega_{7 \times 2 \times 2 \times 7 \times 2 \times 2})$ (dim($\mathcal{B}$) = 327).

(d) Result with base sub-manifold $\mathcal{B} = \mathcal{M}_4(\Omega_{7 \times 2 \times 2 \times 7 \times 2 \times 2})$ (dim($\mathcal{B}$) = 592).

Figure 12: (*Top*) Forward projected data with tensor structure $\mathbb{R}_{\geq 0}^{7 \times 2 \times 2 \times 7 \times 2 \times 2}$. (*Bottom*) Heat map of corresponding $\theta$ values.

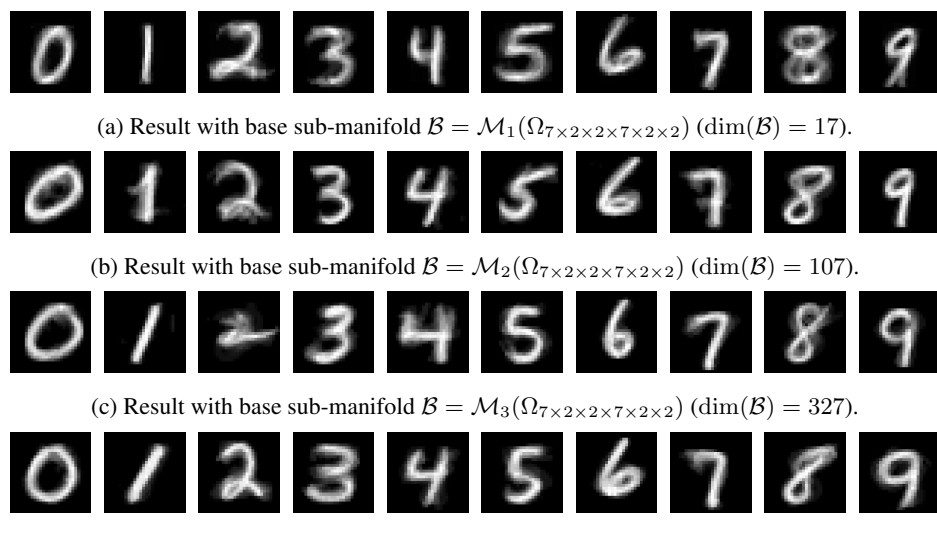

(a) Result with base sub-manifold $\mathcal{B} = \mathcal{M}_1(\Omega_{7\times2\times2\times7\times2\times2})$ ($\dim(\mathcal{B}) = 17$).

(b) Result with base sub-manifold $\mathcal{B} = \mathcal{M}_2(\Omega_{7\times2\times2\times7\times2\times2})$ ($\dim(\mathcal{B}) = 107$).

(c) Result with base sub-manifold $\mathcal{B} = \mathcal{M}_3(\Omega_{7\times2\times2\times7\times2\times2})$ ($\dim(\mathcal{B}) = 327$).

(d) Result with base sub-manifold $\mathcal{B} = \mathcal{M}_4(\Omega_{7\times2\times2\times7\times2\times2})$ ($\dim(\mathcal{B}) = 592$).

Figure 13: Augmented data via Algorithm 4.2 with tensor structure $\mathbb{R}_{\geq0}^{7\times2\times2\times7\times2\times2}$

Similar trends can be found in the case of $\mathbb{R}_{\geq0}^{7\times4\times7\times4}$, as shown in Figures 14 and 15.

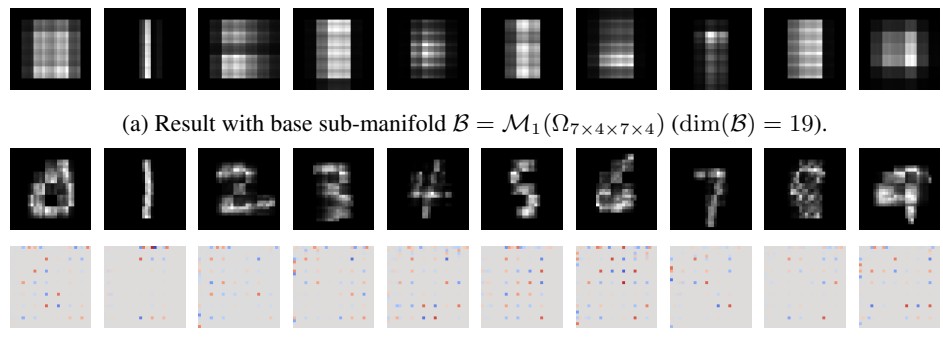

(a) Result with base sub-manifold $\mathcal{B} = \mathcal{M}_1(\Omega_{7\times4\times7\times4})$ ($\dim(\mathcal{B}) = 19$).

(b) Result with base sub-manifold $\mathcal{B} = \mathcal{M}_2(\Omega_{7\times4\times7\times4})$ ($\dim(\mathcal{B}) = 136$).

Figure 14: (*Top*) Forward projected data with tensor structure $\mathbb{R}_{\geq0}^{7\times4\times7\times4}$. (*Bottom*) Heat map of corresponding $\theta$ values.

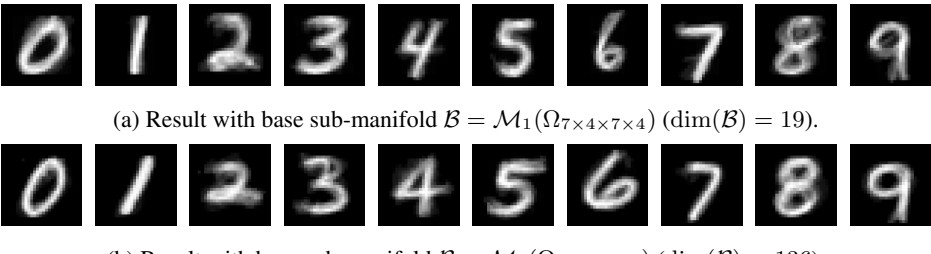

(a) Result with base sub-manifold $\mathcal{B} = \mathcal{M}_1(\Omega_{7\times4\times7\times4})$ ($\dim(\mathcal{B}) = 19$).

(b) Result with base sub-manifold $\mathcal{B} = \mathcal{M}_2(\Omega_{7\times4\times7\times4})$ ($\dim(\mathcal{B}) = 136$).

Figure 15: Augmented data via Algorithm 4.2 with tensor structure $\mathbb{R}_{\geq0}^{7\times4\times7\times4}$.

If we look at the results when using the original matrix structure $\mathbb{R}_{\geq0}^{28\times28}$ (Figures 16 and 17), some interesting comparison can be made. Firstly, if we compare the augmentation results for $\mathcal{B} = \mathcal{M}_1(\Omega_{28\times28})$ (Figure 17) with the finer structures counterparts, e.g., Figure 13(a) for

$\mathcal{M}_1(\Omega_{7\times2\times2\times7\times2\times2})$, one can observe that the results are worse. However, the former requires more dimension $(\dim(\mathcal{M}_1(\Omega_{28\times28})) = 55 > 17 = \dim(\mathcal{M}_1(\Omega_{7\times2\times2\times7\times2\times2})))$ for the base sub-manifold. Similarly, the augmentation results with $\mathcal{B} = \mathcal{M}_1(\Omega_{7\times4\times7\times4})$ (Figure 15(a)) also achieve better performance with a lower base sub-manifold dimension.

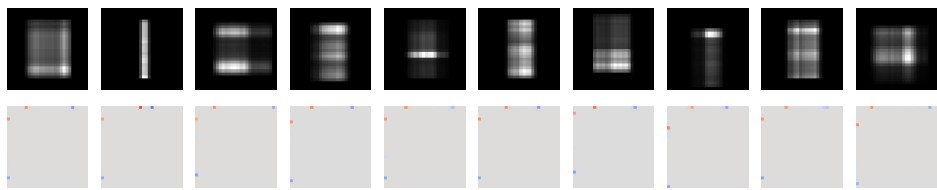

Figure 16: Result with base sub-manifold $\mathcal{B} = \mathcal{M}_1(\Omega_{28\times28})$ $(\dim(\mathcal{B}) = 55)$. (*Top*) Forward projected data with tensor structure $\mathbb{R}_{\geq0}^{28\times28}$. (*Bottom*) Heat map of corresponding $\theta$ values.

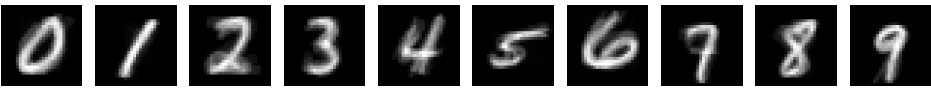

Figure 17: Augmented data via Algorithm 4.2 with tensor structure $\mathbb{R}_{\geq0}^{28\times28}$ and base sub-manifold $\mathcal{B} = \mathcal{M}_1(\Omega_{28\times28})$ $(\dim(\mathcal{B}) = 55)$.

**Remark A.1.** *Many-body approximation is a type of "oblivious" construction for base sub-manifolds that is expected to work well for general positive tensor data. However, alternative choices for $\mathcal{B}$, not necessarily $\mathcal{M}_\ell(\Omega)$ for some $\ell$, could be investigated when specific knowledge about the underlying data is available. Nonetheless, our approach demonstrates non-trivial performance, both in downstream tasks and through visual inspection.*

### A.5 ADDITIONAL DATASET

In addition to Section 5.3, we demonstrate our approach's efficacy on various non-image datasets in this section. In particular, we consider the following UCI datasets:

(a) *Connectionist Bench (Sonar, Mines vs. Rocks)* (Sejnowski & Gorman, 1988) (208 samples, 60 features, 2 classes),

(b) *Taiwanese Bankruptcy Prediction* (Journal, 2020) (6819 samples, 95 features, 2 classes),

(c) *Musk (Version 2)* (Chapman & Jain, 1994) (6497 samples, 12 features, 11 classes).

We summarize the parameters we use for Algorithm 4.2 on each dataset in Table 2. In particular, for each dataset, the training set consists of $80\%$ of the total dataset, and the remaining $20\%$ is used for testing accuracy. Furthermore, the augmented dataset generated also consists of $20\%$ of the training data for each class, similar to the setting in Section 5.3. On the other hand, the autoencoder model trained in each experiment is with hidden-dimension $\dim(\mathcal{B})$ for a fair comparison.

Table 2: Setting of Algorithm 4.2 for different UCI datasets.

| Dataset | Poset $\Omega$ | Base $\mathcal{B}$ | Local Data $\mathcal{D}$ | Bandwidth | $k$ |
|---|---|---|---|---|---|
| (a) | $\Omega_{2\times2\times3\times5}$ | $\mathcal{M}_2(\Omega_{2\times2\times3\times5})$ | $\mathcal{M}_1^*(\Omega_{2\times2\times3\times5})$ | 0.05 | 4 |
| (b) | $\Omega_{5\times19}$ | $\mathcal{M}_1(\Omega_{5\times19})$ | $\mathcal{M}_1^*(\Omega_{5\times19})$ | 0.05 | 8 |
| (c) | $\Omega_{2\times2\times3}$ | $\mathcal{M}_2(\Omega_{2\times2\times3})$ | $\mathcal{M}_1^*(\Omega_{2\times2\times3})$ | 0.1 | 10 |

The results are shown in Table 3. The models perform similarly in most cases, with the fact that augmentation indeed helps with the downstream tasks' performance. We conclude that Algorithm 4.2 is competitive compared to autoencoders regarding the quality of the downstream tasks.

Table 3: Test accuracy of the linear classifier trained on different datasets.

| Dataset | Training Set | | | | |
|---|---|---|---|---|---|
| | Original | Ours | AE | Original+Ours | Original+AE |
| (a) | $60.10 \pm 11.46\%$ | $55.50 \pm 14.92\%$ | $66.30 \pm 10.76\%$ | $75.40 \pm 12.40\%$ | $64.80 \pm 12,85\%$ |
| (b) | $96.90 \pm 4.55\%$ | $81.10 \pm 11.12\%$ | $58.3 \pm 14.04\%$ | $97.30 \pm 4.34\%$ | $97.40 \pm 4.82\%$ |
| (c) | $42.40 \pm 14.96\%$ | $21.80 \pm 11.95\%$ | $20.80 \pm 9.79\%$ | $43.00 \pm 11.19\%$ | $44.10 \pm 14.43\%$ |

