# OpenReview forum: "Pseudo-Non-Linear Data Augmentation via Energy Minimization"
_ICLR.cc/2025/Conference — Submitted to ICLR 2025_

### Official Review · Reviewer_nMW1 · 2024-10-29

**Soundness:** 3
**Presentation:** 2
**Contribution:** 1
**Rating:** 1
**Confidence:** 4

**Summary:**

The paper proposes a data augmentation setup by using a new encoder-decoder approach and then using a simple KDE model to sample from the latent state and use the reconstructed point as additional training data.

**Strengths:**

- The paper is mostly well-written and easy to follow

**Weaknesses:**

Main issues:
- The abstract promises "our approach replaces these non-interpretable transformations with explicit, theoretically grounded ones" but does not deliver any theoretical grounding, and as such this claim is very misleading. In the end, your reconstruction has the same ambiguity problem that it solves using the closest element in some arbitrary norm instead of another arbitrary norm.

- Expanding on the previous point, the paper uses Riemannian geometry concepts but it isn't really clear why they are needed and what is the reasoning. It feels like these ideas were added for extra complexity without any added value. For example in Example 4.2 you start with a positive tensor as input and using your multi-set embedding and poset formulation you end up with an embedding that is simply the normalized flattened tensor.

- Key parts of the algorithm weren't explained. For one you say you project to a flat base sub-manifold, but don't explain how this should be done or how one should identify such a submanifold as these are not trivial conditions.

- Your reconstruction is based on a "local data submanifold" that again isn't explained properly. In the example, you take the nearest neighbors and (if I understand the notations correctly) constrain it to have some of the indices fixed to one of the nearest neighbors. This first of all is a linear subspace so again it raises the question of why do I need the whole Riemannian geometry mechanism. Second, this uses only one of the nearest neighbors so how do you pick the one? Why don't you use the others?

- Experiments are very nonconvincing, they use the very simple MNIST dataset (and even so with only 500 samples), they don't even compare to the basic PCA, and their gain is within the error margins so that the claim that this approach had any positive impact cannot be said with any statistical significance.

**Questions:**

What benefit does the Riemannian framework add theoretically or practically? Couldn't the approach and your experiments be described using only simpler linear algebra?

---

### Official Review · Reviewer_Rdvo · 2024-10-31

**Soundness:** 3
**Presentation:** 2
**Contribution:** 2
**Rating:** 3
**Confidence:** 3

**Summary:**

The paper presents a new generative model based on information geometry principles. The main appeal of the method is that it is interpretable, unlike NN-based models. According to the proposed approach, data is first mapped to posets, then it is embedded into the statistical manifold having coordinate systems defined by natural and expectation parameters of exponential distributions. On this new representation encoding and decoding can be done. The authors suggest a projection scheme for encoding points, a method based on KDE for sampling novel points on the embedded submanifold, and a decoding scheme to project points backward to the manifold. The authors demonstrate their approach on MNIST and three UCI datasets.

**Strengths:**

* As far as I know, the proposed approach is very original and novel.
* It does seem to provide some measure of interpretability (at least on MNIST, it wasn't demonstrated on the UCI datasets).
* Figures 1 and 2 are great and help to get a better intuition.

**Weaknesses:**

Although novel and interesting, in my opinion, the paper is not ready for publication at ICLR and requires substantial work. Specifically,
* Clarity of exposition:
  * Information geometry is relatively a small niche in the field of ML. It is not a bad thing by any means, but it requires adding a substantial background on the field and the main concepts, even concepts that arrive from standard differential and Riemannian geometry. I do not think that simply referring to Amari (2016) as the authors did is nearly enough.
  * I believe that the paper needs to be restructured in a different way. Example 4.2 is great. I think that the authors should use it as a running example throughout the paper (from the start of section 4) and demonstrate each element of the method using it.
  * In my opinion the paper will be much clearer if the authors use a concrete example (i.e., vectors with numbers) to demonstrate every step of the algorithm (mapping to posets, mapping to the statistical manifold, etc.). Preferably it should be done with the running example.

* I think it would help to clarify the design choices in the paper and discuss their advantages and limitations. For instance, when is modeling with postes or using the many-body tensor approximation challenging, and why? In addition, perhaps the authors should clarify if there are alternatives to these choices and why not choose them.

* Experiments:
  * Evaluating the model on MNIST and UCI datasets is not enough. Even adding CIFAR-10 or data from other modalities (e.g. audio) will greatly improve the empirical evaluation part.
  * Perhaps I do not understand something, but I do not see an improvement in the generated augmented data when increasing the dimension of the submanifold. In addition, for all sizes, the image quality is not good. Do you think it can be made better?
  *  The authors state that AEs are overfitting the training set (Lines 430-431), yet to me it seems that this approach also overfits as the generated images look very similar to the closest images on the training data in all figures. Perhaps the improvements in test accuracy (Table 1) arise from the blurring effect (which acts as a form of standard data augmentation)? Further empirical evaluation will help clarify this ambiguity.

* Minor:
  * In line 29, the paper by Kingma & Welling, is from 2014.
  * In line 395, did you mean 100 images per digit?

**Questions:**

No.

---

### Official Review · Reviewer_1jmw · 2024-11-01

**Soundness:** 2
**Presentation:** 2
**Contribution:** 3
**Rating:** 5
**Confidence:** 3

**Summary:**

The authors propose a novel, interpretable data augmentation approach for structured data, that perturbs data within a dimensionality-reduced space.  Specifically,
- This space is constructed by i) embedding structured data into a *statistical manifold* (*i.e.*, discrete probability distributions) via an explicit mapping $\phi$ using an energy-based model (built upon the log-linear model on posets), and ii) projecting the embedded data into a low-dimensional sub-manifold.
- The perturbation is performed by fitting a kernel density estimator to the data distribution in this sub-manifold and sampling from it.
- The perturbed data is then *projected back* into the original space through a *backward projection* algorithm. This algorithm creates a local mapping around each perturbed point using the embeddings of the nearest known training points.

The proposed data augmentation is evaluated on MNIST by training a linear classifier, and compared with using an autoencoder model ($2+2$ layers) as data augmentation.

**Strengths:**

- The paper tackles a relevant research question through an elaborate and original angle. The approach is well-motivated and the paper is well written
- The proposed augmentation method is an interpretable generative approach that stands out from black-box generative models.
- The approach is applicable to any kind of structured data, which gives it a lot of potential.

**Weaknesses:**

**Method section**

I am quite unfamiliar with the theory underlying Sections 3 and 4.1, and it is very difficult to parse these sections without a good knowledge of the domain.
I believe that the paper would be more "accessible" if Section 4 started with a more global outline, where the different steps of the augmentation approach are mentioned: 1. Embedding on a statistical manifold, 2. Projection into a sub-manifold, 3. Perturbing the data, 4. Defining a backward projection.
It would be good to guide the reader on the contents of the different subsections, such that the fact that section 4.1 focuses on step 1. One should not have to read the entire Section 4 to understand its global structure.

Illustrative figures such as Figure 2 should be introduced early in Section 4 to directly provide a broad overview of the whole augmentation process, with a caption helping the figure to be self-contained.


**Experimental validation**

My main concern regards the empirical validation of the proposed augmentation method.
The visual and quantitative evaluations are solely based on MNIST. This is very far from demonstrating the potential of the proposed approach.
The performance obtained using a simple linear classifier trained on $28 \times 28$ digit images will not give a clue on whether the augmentation helps in real scenarios with **natural images of higher resolution** and **beyond linear classification**. The paper should also evaluate whether the proposed augmentation **complements or outperforms standard data augmentation methods**.

Even a small-scale evaluation would be valuable, like training a ResNet-18 on a small dataset such as CUB, with appropriate data augmentation (e.g., color jittering, random flipping, random resizing and cropping), and seeing whether adding the proposed augmentation improves results. The proposed method could also be compared to simple, image-level perturbations such as Gaussian noise or blurring. Along with quantitative evaluations, visualizations on data beyond MNIST would also be appreciated.

One of the strengths of the proposed approach being its applicability to any kind of structured data, I would have liked to see evaluations on data structures beyond 2D images.

**Questions:**

**Suggestions** – Please refer to the Weaknesses section:
- Reorganize the method section to better guide the reader, starting with a global overview of the proposed approach, illustrated with a figure.
- Conduct a thorough experimental validation on **real, higher-resolution datasets, beyond simple linear classification, and integrating standard data augmentations.**
- Expand the evaluation to include structured data types beyond 2D images

---

### Official Review · Reviewer_wZ51 · 2024-11-03

**Soundness:** 3
**Presentation:** 3
**Contribution:** 2
**Rating:** 3
**Confidence:** 3

**Summary:**

The paper uses poset (partially-ordered set) to capture structures in the data.
Specifically, given a dataset, the data is modeled by a real-valued poset, where the partial ordering is chosen based on the data.

Then, data augmentation is performed in 5 steps:
1. map the poset to a statistical manifold $\mathcal{S}$ using an embedding function $\varphi$ (which can be nonlinear);
2. project to a lower-dimensional manifold $\mathcal{B}$ for dimension reduction;
3. sample from $\mathcal{B}$ (the distribution is estimated by e.g. a kernel density estimator);
4. backward project from $\mathcal{B}$ to $\mathcal{S}$; specifically, given a point $w^* \in \mathcal{B}$, let $w_1, \cdots, w_k \subset \mathcal{B}$ be the $k$-nearest neighbors of $w^*$ and let $z_1', \cdots, z_k' \subset \mathcal{S}$ be the corresponding pre-images, then the backward projection will map $w^*$ to a projection on the local manifold defined by $z_1', \cdots, z_k'$.
5. map to the space of the original data using the inverse of the embedding function, i.e. $\varphi^{-1}$.

The above method is _pseudo-non-linear_, in the sense that it is an interplay between linear constraints on the manifold (from log-linear energy models), and nonlinear mapping from the original data to the manifold.

The paper then empirically verifies the effectiveness of this data augmentation method. Using data augmented by the proposed method improves the MNIST accuracy from 81.8% to 83.4%. In comparison, data augmented using an autoencoder (AE) improves the accuracy to 82.7%.

**Strengths:**

- The proposed method has a motivation from information geometry.
- The paper provides the necessary background.

**Weaknesses:**

- The proposed method provides limited empirical gain. The proposed method outperforms AE by 0.7% accuracy on MNIST and 6.6% gain on Connectionist Bench, but performs worse than AE on the other two datasets (Taiwanese Bankruptcy Prediction and Musk).
- The experiments are limited to small-scale datasets (with <10k samples and <1k dimension), and it's unclear whether the proposed method is scalable (for computational reasons) to more complex datasets.
- There's limited quantitative understanding on how much the data augmentation helps. For example, how does the performance change as a function of the number of augmented samples?
- The proposed framework requires partial ordering, and hence cannot capture invariant or equivariant structures, which the paper also mentions.

**Questions:**

- Could you comment on the practical applicability (e.g. performance gain, computational cost, scalability) of the method?
- Line 430, comparing the proposed method and AE: the paper says that the proposed method gives a blurred effect; could you clarify why this is desirable?
  - Note also that blurring can also be obtained by using a VAE. Moreover, the paper seems to think of the blurred effect as a benefit that differentiate the proposed method from AE, whereas blurring is often considered an undesired behavior of VAE.

---

### Meta-Review · Area_Chair_hkRT · 2024-12-08

**Metareview:**

This paper explores data augmentation methods based on energy-based modelling.

Reviewers raised many concerns with the work, including that there was a lack of theoretical results contrary to the claim the work was theoretically grounded, that the methods section was not clearly laid out and summarized, there were no figures that help visualize the method, and that the experimental results were small-scale, did not show significant improvements, and did not consider ablations to provide quantitative understanding of the method

The authors did not respond to any of the reviews within the discussion period. Due to the author’s lack of engagement in the peer review process, I am recommending rejection.

**Additional Comments On Reviewer Discussion:**

No discussion or changes occurred during the rebuttal period as the authors did not respond.

---

### Decision · Program_Chairs · 2025-01-22

Reject